# Progesterone Signaling and Uterine Fibroid Pathogenesis; Molecular Mechanisms and Potential Therapeutics

**DOI:** 10.3390/cells12081117

**Published:** 2023-04-09

**Authors:** Mohamed Ali, Michał Ciebiera, Somayeh Vafaei, Samar Alkhrait, Hsin-Yuan Chen, Yi-Fen Chiang, Ko-Chieh Huang, Stepan Feduniw, Shih-Min Hsia, Ayman Al-Hendy

**Affiliations:** 1Department of Obstetrics and Gynecology, University of Chicago, Chicago, IL 60637, USA; mohamed.ali@bsd.uchicago.edu (M.A.);; 2Clinical Pharmacy Department, Faculty of Pharmacy, Ain Shams University, Cairo 11566, Egypt; 3Second Department of Obstetrics and Gynecology, Center of Postgraduate Medical Education, 00-189 Warsaw, Poland; 4School of Nutrition and Health Sciences, College of Nutrition, Taipei Medical University, Taipei 11031, Taiwanbryanhsia@tmu.edu.tw (S.-M.H.); 5Department of Gynecology, University of Zurich, 8091 Zurich, Switzerland

**Keywords:** progesterone, uterine fibroids, leiomyoma, SPRM, ulipristal, natural compounds

## Abstract

Uterine fibroids (UFs) are the most important benign neoplastic threat to women’s health worldwide, with a prevalence of up to 80% in premenopausal women, and can cause heavy menstrual bleeding, pain, and infertility. Progesterone signaling plays a crucial role in the development and growth of UFs. Progesterone promotes the proliferation of UF cells by activating several signaling pathways genetically and epigenetically. In this review article, we reviewed the literature covering progesterone signaling in UF pathogenesis and further discussed the therapeutic potential of compounds that modulate progesterone signaling against UFs, including selective progesterone receptor modulator (SPRM) drugs and natural compounds. Further studies are needed to confirm the safety of SPRMs as well as their exact molecular mechanisms. The consumption of natural compounds as a potential anti-UFs treatment seems promising, since these compounds can be used on a long-term basis—especially for women pursuing concurrent pregnancy, unlike SPRMs. However, further clinical trials are needed to confirm their effectiveness.

## 1. Introduction

Uterine fibroids (UFs) are the most common benign tumor of the female genital tract that develops in the muscular wall of the uterus. Recent findings show that stem cells, growth factors, genetic and epigenetic factors, ovarian steroid hormones, cytokines and chemokines, and extracellular matrix (ECM) components are the critical factors involved in the development and growth of UFs [1,2,3,4,5,6,7,8]. The UF pathogenesis model invokes the genetic transformation of a single myometrial stem cell (MMSC) into a tumor-initiating cell (UFSC) that seeds and sustains clonal tumor growth. This is characterized by an increase in cell size and number as well as abundant ECM production, under the influence of endocrine, autocrine, and paracrine growth factors and hormone receptor signaling [4,9,10,11,12,13,14,15]. Fibroids also elicit mechanotransduction changes that result in decreased uterine wall contractility and increased myometrium rigidity, which affect normal biological uterine functions such as menstrual bleeding, receptivity, and implantation [16]. UFs affect the normal endometrium by modifying the vascular architecture, impairing normal contractility, and altering the production of angiogenic factors such as vascular endothelial growth factor (VEGF), cytokine tumor necrosis factor alpha (TNF-a), chemokines, and growth factors such as transforming growth factor beta (TGF-β), basic fibroblast growth factor (BFGF), epidermal growth factor (EGF), platelet-derived growth factor (PDGF) [16,17,18]. In addition, several mechanisms have been proposed to explain the anti-fertility effects of fibroids. These include the simple physical impedance by impairing and/or obstructing the transport of gametes or embryos, hindering implantation, acceleration of myometrial peristalsis in the mid-luteal period, and/or increasing the milieu prolactin and aromatase levels. However, the most compelling evidence is likely related to the negative impact of fibroid secretome on endometrial receptivity [19]. One of these mechanisms discussed bone morphogenetic protein 2 (BMP-2) function, a critical factor in the process of embryo implantation [20,21,22,23,24] in relation to UFs. BMP-2 is a growth factor that belongs to the TGF-β superfamily. It regulates cell proliferation and differentiation. Conditional ablation of BMP-2 in murine endometrium results in complete infertility [20]. In addition, impairment in BMP action was shown in the endometrial cells of women with UF uteri due to over-secretion of TGF-β [25]. The study showed that the blockade of TGF-β prevents repression of BMP-2 receptors and restores BMP-2-stimulated expression of homeobox A10 (HOXA10) and leukemia inhibitory factor (LIF) in the endometrial stromal cells. HOXA10 is a transcription factor known to be regulated by estrogen and progesterone and is essential for endometrial receptivity [26,27], while LIF is an important endometrial biomarker for uterine implantation capacity [28,29,30]. It has been documented that women suffering from UFs have decreased endometrial HOXA10 [25] and LIF [31] expression in the endometrium [2,3,4,5,6,7,8].

Although the leading causes of fibroids are unknown, accumulating epidemiological, clinical, and experimental evidence supports the pivotal role of ovarian steroid hormones in the growth and pathogenesis of UFs [32]. As a steroid hormone, progesterone is responsible for preparing/maintaining the endometrium for pregnancy and controlling the menstrual cycle [33]. Importantly, progesterone action is complex and involves interaction with receptors, transcription factors, kinase proteins, growth factors, and numerous autocrine and paracrine factors. Previous studies suggested that estrogen predominantly increases tissue sensitivity to progesterone by expanding the availability of progesterone receptors (PR) [34]. Conflicting results remain regarding the stimulatory or inhibitory effects of progesterone in UF development, as the mechanisms of progesterone action are not fully understood. Moreover, the promising results of utilizing selective progesterone receptor modulators (SPRM) in controlling UF-associated symptoms such as bleeding as well as tumor shrinkage, further supports the role progesterone plays in UF pathogenesis [32]. In this article, we reviewed the literature covering progesterone signaling in UF pathogenesis and discussed the potential of anti-UF therapeutics via modulation of progesterone signaling, including pharmaceutical drugs as well as promising natural compounds.

## 2. Progesterone Signaling in Uterine Fibroid Pathogenesis

Generally, progesterone modulates physiological processes through two actions, which are classical and nonclassical. Classical actions are mediated by nuclear progesterone receptors (nPRs). These nPRs have two well-described isoforms such as PR-A and PR-B, which were shown to have higher expression in UFs tissue compared with adjacent myometrium [35]. Non-classical progesterone actions are mediated by non-nuclear PR, which belongs to the membrane-associated progesterone receptor (MAPR) family. This includes progesterone receptor membrane component 1 (PGRMC1) and PGRMC2, which are expressed in several reproductive organs including the placenta and ovary [36]. So far, progesterone binding has only been tested for PGRMC1, although the specific binding sites have not yet been identified. In addition, PGRMC2 was shown to play a role in uterine decidualization [37].

Disrupted progesterone signaling is causally linked to many reproductive diseases that result in subfertility or infertility, including UF [38]. Progesterone signaling, via its receptor, can affect several pathways and interact with different molecules, which in turn stimulates cells towards UF phenotype through the following functions:

### 2.1. Proliferation Induction

Studies demonstrated that progesterone signaling promotes the growth and proliferation of fibroid cells through increasing proliferating cell nuclear antigen (PCNA) expression, which increases both the number of fibroid cells and the size of fibroid tumors [39]. This is thought to be achieved by increasing the expression of growth-promoting genes and decreasing the expression of growth-inhibiting genes; for example, activating the mitogen-activated protein kinase (MAPK) pathway [40]. In addition, progesterone induced expression of several WNT ligands, with the subsequent induction of β-catenin nuclear translocation and transcriptional activity of its heterodimeric partner T-cell factor and their target gene *AXIN2*, leads to the proliferation of fibroid side population (SP) cells [14]. The side population cell phenotype was first described in bone marrow, where a somatic stem cell population was identified based on its ability to extrude the DNA-binding dye Hoechst 33342, a phenomenon that is associated with the expression of the ATP-binding cassette transporter G2 [41]. The SP phenotype is thought to be a universal marker of somatic stem cells and has been used to isolate them from many adult tissues such as the myometrium, endometrium, and mammary glands [11,42,43,44]. Several groups reported that SP cells from human UFs exhibit key features of the tumor-initiating cells [4,45]. Moreover, the TGF-β pathway is well known to regulate cell growth and division, and high levels of TGF-β have been associated with an increased risk of fibroid development [46,47]. Progesterone has been shown to increase the production of TGF-β, which in turn stimulates UFs’ progression [47].

### 2.2. Apoptosis Suppression

Apoptosis, or programmed cell death, is a mechanism by which abnormal or damaged cells are eliminated from the body. Progesterone has been shown to suppress apoptosis in fibroid cells, which may contribute more to the growth of UFs via B-cell lymphoma- (Bcl-) 2 protein expression induction [48] than myometrium [49]. In fact, progesterone upregulates bcl-2 mRNA, while estrogen downregulates bcl-2 protein [50]. Furthermore, Yin et al. found that liganded PR binds to the bcl-2 promoter and enhances bcl-2 transcription in primary cultured UF cells [51]. SPRMs like asoprisnil decreased antiapoptotic bcl-2, with a corresponding increase in TUNEL staining, cleaved caspase 3, and cleaved PARP supporting a role for PR in preventing apoptosis in these cells [52]. The progestin, R5020, can rapidly activate the AKT pathway with subsequent cell proliferation induction and apoptosis suppression of UFs [53].

### 2.3. Regulation of Cytokine Production

Cytokines are signaling molecules that play a significant role in the regulation of fibroid growth. Progesterone has been shown to modulate the production of cytokines such as TNF-α and IL-6 [54]. In addition, progesterone upregulates EGF, which is mitogenic and stimulates TGF- β3 expression; meanwhile, it downregulates insulin-like growth factor (IGF)-I expression through PR-B, while PR-A appears to inhibit this function [55]. On the other hand, the receptor activator of NF-κB (*RANKL*) was recently identified as a novel progesterone/PR responsive gene that plays a vital role in promoting the growth of UFs by activating cyclin D1 [56]. Furthermore, RANKL signaling inhibitor RANK-Fc inhibited estrogen- and progesterone-induced UFs regeneration *in vivo*, which may provide a potential alternative therapeutic strategy [57].

### 2.4. DNA-Methylation

The findings of Liu et al. suggested that DNA methylation and MED12 mutation together constitute a complex regulatory network that affects progesterone/PR-mediated *RANKL* gene expression, with an essential role in activating stem cell proliferation and fibroid tumor development [56]. Neonatal exposure to glyphosate-based herbicides (GBHs) altered the expression and induced hypermethylation of the *Hoxa10* gene in uterine tissue in early life and increased the E2/P4 ratio serum level in middle age. This indicates estrogen dominance and is considered to be a risk factor for hormonal carcinogenesis [58]. By dissecting the complex interactions between progesterone action and DNA methylation during UF stem cell differentiation, new insights into the mechanisms behind the epigenomic regulation of hormone-dependent tumor growth will be provided [59].

### 2.5. Angiogenesis

Since the formation of new blood vessels is vital for the delivery of nutrients and oxygen to the growing tumors [60], progesterone may promote angiogenesis through the upregulation of VEGF which stimulates the growth of blood vessels [61]. Vascular permeability is triggered by secreted permeability factors such as VEGF that are controlled by PR [62]. Kim et al. indicated that VEGF-A regulated by progesterone governs uterine angiogenesis and vascular remodeling during pregnancy [61]. Moreover, the UF’s abnormal vasculature, together with its aberrant hypoxic and angiogenic response, may make it especially vulnerable to the disruption of its vascular supply, a feature which could be exploited for treatment [63]. Treatments include UPA-inhibited cell proliferation and angiogenesis in UFs [32].

### 2.6. ECM

ECM is an intricate network composed of an array of multidomain macromolecules organized in a cell/tissue-specific manner that provide structural support to cells [64]. In fibroids, the ECM is abnormal and may contribute to the growth and maintenance of the tumors. Additionally, progesterone signaling leads to the upregulation of matrix metalloproteinases (MMPs) which are involved in ECM remodeling [65]. Moreover, collagen is a major component of the ECM in UFs and is therefore essential for their growth and stability, and progesterone has been shown to increase collagen synthesis in fibroid cells [9]. In UF xenografts, it was proved that excessive ECM production in UFs is regulated by steroid hormones via the downregulation of miR-29b [66]. MMSCs formed organoids and showed responsiveness to the core regulatory hormone estrogen with an overall increase in size. Interestingly, MMSC and UFSC developed organoids showed a differential gene expression profile in response to progesterone treatment in terms of progesterone-responsive genes expression [67].

All these molecular mechanisms of action demonstrate the complex and multi-faceted role of progesterone in the regulation of UFs (Figure 1). Further research is needed on the underlying mechanisms and developing effective treatments for fibroids. Thus, given the role of progesterone in the growth and maintenance of UFs, targeting progesterone signaling may be an effective approach to treating these tumors, including SPRMs and progesterone receptor antagonists [68]. SPRMs are synthetic steroid ligands targeting PR, and possess tissue-selective effects comprised of mixed agonist and antagonist activities [69]. Their investigated indications include emergency contraception (EC), termination of pregnancy, premenstrual syndrome, and assisted reproduction. Additionally, SPRMs affect endometrial and fibroid cells directly; therefore, they were investigated against UF, abnormal uterine bleeding (AUB), dysmenorrhea, endometriosis, and breast cancer prevention [70].

## 3. Selective Progesterone Receptor Modulator and Uterine Fibroids

A Cochrane review was conducted in 2017 using data from 11 randomized controlled trials, and included 1215 study participants. It consisted of five studies about mifepristone, four about ulipristal acetate (UPA), and two about asoprisnil [71]. Fibroid-related symptoms were measured using the quality of life (QoL) questionnaire, and both menstrual bleeding and pelvic pain were assessed. Moreover, the volume of fibroids was evaluated. In comparison of SPRM with placebo, the symptoms severity scale (SS-QoL—range 0–100) assessing bleeding, abdominal pressure, urinary frequency, and fatigue reported an average of 20 points reduction following SPRM utilization. Briefly, relief of bleeding, decreased menstrual blood loss, and amenorrhea were significantly more often observed in participants receiving SPRM than placebo. However, there were no statistical changes in pelvic pain during the studies. All studies, compared to placebo, showed a crucial decrease in fibroid and uterus volume after using the SPRM. Unlike SPRM versus leuprolide acetate (LA), no significant differences in symptoms between treatment methods were observed, while the uterine and fibroids size was decreased in SPRM studies [71]. Notably, LA was, back then, the only approved pharmacological treatment for presurgical hematologic improvement in women with UFs, and was considered the only standard drug to explore any emerging drug efficacy [72,73].

### 3.1. Mifepristone

In a study by Murphy et al., mifepristone was found to be effective in treating symptomatic fibroids in 10 patients, with a significant decrease in lesion dimensions (49.0 ± 9.2%) observed after 3 months of therapy with 50 and 25 mg daily [74]. A recent systematic review from 2021 reviewed 11 studies with 902 participants from China [75]. The usefulness of a traditional Chinese medication, Xuefu Zhuyu decoction, combined with mifepristone, was assessed in treating fibroids. A significant reduction in fibroid volume and decreased levels of estradiol, luteinizing hormone (LH), follicle-stimulating hormone (FSH), and progesterone was observed in the combination treatment group compared to mifepristone alone [75].

Another meta-analysis also showed that mifepristone was able to lower uterine and fibroid volume considerably and decrease related symptoms, including heavy menstrual bleeding, dysmenorrhea, pelvic pain, and anemia. The optimal dose of between 2.5 and 5 mg/day of mifepristone for a period of 3–6 months was established. Notably, the authors pointed out that special attention should be paid to the long-term observation of endometrial hyperplasia [76].

In phase III of a placebo-controlled study (NCT00133705), 5 mg/day of mifepristone influenced menstrual bleeding, quality of life, pain, and uterine size in 42 women with symptomatic UFs [77]. After 26 weeks of observation, uterine size was reduced by an average of 47%, and amenorrhea was achieved in 41% of patients, with an improvement in blood count rates in the mifepristone group [77]. In comparison, between 2.5 mg/day and 5 mg/day of mifepristone (NCT01786226), there were 78 and 94% of women who experienced amenorrhea during the 9-month observation period, respectively. Moreover, fibroid volume was decreased by 27.9 and 45.5% after 3 months of observation. However, there was no difference in quality of life improvement between the two mifepristone studied groups [78]. As for side effects, two compared doses (2.5 vs. 5 mg/day) resulted in 9 and 16% reported hot flashes, while 2 and 4% suffered nausea and vomiting, respectively. The liver enzymes were elevated in 13 and 7% of the women, respectively. No evidence of endometrial hyperplasia was observed in both studies [77,78].

### 3.2. Ulipristal Acetate (UPA)

Ulipristal acetate (UPA) is the most known and tested SPRM [76]. The effectiveness of UPA in bleeding reduction is comparable to gonadotrophin-releasing hormone (GnRH) agonists, and menopausal symptoms are less likely because of the influence on the estradiol level. Several phase III clinical trials assessed the UPA effect on women with fibroids. The PEARL I trial (NCT00755755) explored the effectiveness of 5 and 10 mg/day of UPA compared to placebo in 13 weeks of observation [79]. Decreased blood loss was shown in 19, 91 and 92% of women after administration of placebo, 5 and 10 mg/day UPA, respectively. Amenorrhea was present in 6, 73, and 82% of patients receiving placebo, 5 and 10 mg/day UPA, respectively. Fibroids volume was reduced by 3, 21 and 12% after placebo, 5 and 10 mg/day of UPA administration. No differences were observed in side effects or in glucose, estradiol, corticotropin, or prolactin levels [79].

In the PEARL II trial (NCT00740831), administration of 5 and 10 mg of UPA was compared to 3.75 mg of leuprolide acetate in 3 months of observation [73]. No differences were found in the reduction of bleeding, which oscillated around 90% across the groups. A slightly lower decrease in fibroid volume was observed, with 36, 42 and 53% of patients receiving 5, 10 mg/day of UPA and leuprolide acetate, respectively. Adverse events, such as hot flashes, were significantly higher in a group of patients after leuprolide acetate administration [73]. The PEARL III extension study (NCT01252069) assessed the safety of 10 mg/day of ulipristal acetate administration, followed by either norethisterone acetate (NETA) or placebo, where the patient received repeated intermittent (1–4 times) open-label UPA courses. Each was followed by randomized double-blind NETA or placebo, to explore any effect on the reversibility of PAEC or timing and the magnitude of the next menstruation off-treatment. The off-treatment period between each UPA course included one menstrual bleed and the beginning of a second bleed. Amenorrhea was present in 80–90% of cases, while severe treatment-emergent adverse events (TEAEs) (headache, nasopharyngitis, and abdominal pain) were reported in eight women (3.8%). PR modulator associated with endometrial changes (PAECs) were reported in 11–26%, but this change was in almost all cases reduced, and no malignant changes were found [80]. The PEARL IV trial (NCT01629563) assessed the efficacy of 5, 10 mg/day of UPA in 12 weeks of observation [81]. Similar outcomes were observed in PEARL I-III studies in amenorrhea, vaginal bleeding, and fibro volume reduction. The pain relief measured on the median visual analogue scale pain scores was reduced from 40 to 6, independent of the UPA doses. QoL was also significantly improved independently of the doses of UPA, and no crucial adverse effects were present [81].

The PREMYA trial (NCT01635452) in Germany, France, UK, Romania, Portugal, Sweden, Poland, Hungary, Slovenia, and Austria assessed the safety, efficacy, and QoL after UPA administration in 15 months of observation. Around 40% of the included participants required surgery. Even after the end of treatment, pain and quality of life were better [82]. In the VENUS I trial (NCT02147197), 5, 10 mg/day of UPA versus placebo in 12 weeks of intervention and 12-week drug-free observation was examined in the US [83]. Amenorrhea was present in 1.8, 47.2 and 58.3% of patients receiving placebo, the 5 and 10 mg/day of UPA, respectively. No significant fibroid volume decrease was observed, with a 7% volume increase in the placebo group. The VENUS II trial (NCT02147158) showed similar outcomes that were observed in VENUS I in amenorrhea, vaginal bleeding, and fibroid volume reduction. No significant differences in adverse effects or PAECs were shown [84]. ClinicalTrials.gov also lists some phase IV trials (NCT02361879, NCT02357563, NCT02288130, NCT02601196, NCT02825719, and NCT03421639).

Chronic use of these agents may be associated with abnormal liver enzymes and, rarely, hepatic failure; however, the underlying mechanism for such induced liver toxicity is still not fully understood [85]. Some years ago, it was described that the use of UPA triggered serious cases of liver injury, with some even leading to liver transplantation. These cases prompted the European Medicines Agency (EMA) to restrict the use of UPA. One explanation for UPA-induced liver injury may be attributed to having common structural features with other identified hepatotoxic drugs, which affect lipophilicity and cause extensive hepatic metabolism and long half-life pharmacokinetics [86,87]. Nonetheless, high-quality data is needed to draw clear conclusions regarding UPA safety.

### 3.3. Asoprisnil

Another SPRM is asoprisnil, which is currently assessed in two placebo-controlled Phase III randomized trials (NCT00152269 and NCT00160381) [88]. Upon comparing placebo to 10 mg/day and 25 mg/day of asoprisnil in 12 months of observation, amenorrhea was observed in up to 12, 78, and 93% of women, respectively. The fibroid volume decreased by 48, 63% after administration of 10 mg/day and 25 mg/day of asoprisnil, and increased by 16% in the placebo group. Anemia rates decreased after SPRM, which influenced the QoL of the included patients. The endometrium was about 2 mm thicker after 12 months of asoprisnil treatment. After 6 months of treatment with 10 mg/day of asoprisnil, hyperplasia without atypia was found more often in the treatment group. After 9 months with 25 mg/day, one low-grade endometrial adenosarcoma was found [88].

### 3.4. Telapristone Acetate

Telapristone acetate was assessed as 12.5, 25, or 50 mg/day dose [89,90]. In 30 women, a 10.6%, 32.6%, and 40.3% reduction in volume was observed after 3 months, respectively, compared to LA (32.6% reduction) and placebo (10.6% increase). The symptoms were also decreased in the study groups [89]. Moreover, telapristone acetate was assessed in several studies in the phase III trials (NCT00702702, NCT00737282, NCT01069120, NCT00735553, NCT00683917, NCT00785356, NCT00853567 and NCT00874302). The results of these studies have not been published yet, and some phase trials have been discontinued because of concerns about patient safety [68].

### 3.5. Vilaprisan

Vilaprisan is the most recently developed SPRM, and was tested in regimens of 0.5 mg/day, 1 mg/day, 2 mg/day, and 4 mg/day as compared to placebo in a placebo-controlled phase II trial called ASTEROID 1 (NCT02131662) [91]. Amenorrhea status, defined as no scheduled or unscheduled bleeding/spotting after the end of the initial bleeding episode until the end of treatment, was achieved in 87–92% of patients after administration of ≥1 mg of vilaprisan, compared to 9% of patients who received a placebo. The mean reduction of fibroid volume was 15–41% after vilaprisan treatment in comparison to a 5% decrease in the placebo group. Side effects (ovarian cyst, headache, and hot flashes) were present in placebo and intervention groups without a significant difference. In early-phase clinical trials, vilaprisan was shown to be safe and effective in ameliorating fibroid-related symptoms, especially in abnormal or excessive uterine bleeding and in volume reduction [92]. Further investigations about safety and efficacy were planned in multiple phase III studies, such as ASTEROID 4 (NCT03400956), ASTEROID 5 (NCT03240523), ASTEROID 6 (NCT03194646), ASTEROID 7 (NCT03699176), and ASTEROID 8 (NCT03476928). The problems with this SPRM were likely due to cases of toxicity in rodent models. More data about the safety of vilaprisan is urgently needed to draw definitive conclusions on its future direction [93].

## 4. Natural Compounds That Can Benefit against Uterine Fibroids through Modulating Progesterone Signaling

Phytoprogestins, plant-derived compounds, are progesterone antagonists and are often used to combat gynecological diseases that can be stimulated by steroid hormones, among which apigenin, kaempferol, luteolin, and naringenin are particularly common [94]. Several studies have suggested that these natural compounds are capable of modulating steroid receptor expression and the release of progesterone in the uterus. For example, Toh et al. demonstrated that kaempferol blocked genistein-induced proliferation in the uterine luminal epithelial cells in ovariectomized rats, and decreased genistein-induced expression of the progesterone receptor protein in human endometrial stromal cells (HESC) [95]. In addition, Dean et al. demonstrated that apigenin abrogated the effect of genistein on uterine epithelial height and induced Hand2 protein expression in the uterus of ovariectomized rats, but did not alter levels of the progesterone receptor [96]. Unfortunately, the above mentioned natural compounds have yet to be explored for UFs. However, several studies have been conducted to explore other multifunctional natural compounds that can regress UFs via modulating progesterone signaling, such as berberine, isoliquiritigenin, coix seed extract, vitamin D, epigallocatechin gallate (EGCG), Curcumae Rhizoma-Sparganii Rhizoma (CRSR) and Curcumae rhizomes, which we will briefly review in the next chapter.

### 4.1. Berberine

Berberine, a natural alkaloid isolated from Berberis species, has a variety of biological activities [97], including improving postmenopausal anxiety [98] and endocrine–metabolic disorder in women [99], as well as possessing anti-tumor effects. In a study on UFs, berberine treatment significantly decreased EKER rat leiomyoma cell line (ELT-3) viability, and reduced the tumor volume along with several UF phenotype related genes such as (COX2 and PTTG1) in ELT3 xenograft tumors [100]. In addition, a previous study showed that berberine treatment (50 μM) was able to block progesterone (100 nM)-induced cell proliferation and apoptosis in human leiomyoma cell lines (HuLM) [101], which suggests that berberine may have anti-progestin effects in UFs. These results indicated that berberine may inhibit fibroid cell growth by modulating progesterone’s effect.

### 4.2. Isoliquiritigenin

Isoliquiritigenin, one of the bioactive components of licorice, has the potential to be used as an anti-cancer agent. As a phytoestrogen, isoliquiritigenin is thought to interact with estrogen receptors and affect the ability of antral follicles to synthesize estradiol. Furthermore, isoliquiritigenin at 100 μM also reduced progesterone levels in ovarian antral follicles [102]. Lin et al. also confirmed the effect of isoliquiritigenin on E2-induced mouse uterine myometrium growth. Although the results showed a decreasing trend in the serum P4 level by co-treating with E2 and isoliquiritigenin at a high dose (5 mg/kg), the effect is still better than the current drug tamoxifen [103]. These results indicated that isoliquiritigenin can interfere with the synthesis of both estradiol and progesterone.

### 4.3. Adlay Extracts

Adlay (Coxi lacryma-jobi L. var. ma-yuen Stapf.) has been used in in vitro experiments to evaluate the viability of UFs cells, and the results showed that adlay sub-fractions were able to inhibit the proliferation of ELT3 and primary UF cells [104]. Based on this phenomenon, in recent years a study has further explored the in vivo experiment of UFs [105]. It demonstrated that the diethylstilbestrol/medroxyprogesterone 17-acetate (DES/MPA)-increased uterine myometrium layer and serum levels of progesterone were significantly reduced by the treatment of ethyl acetate (ea) fraction of adlay hull extract (AHE) (AHE-ea), or its active compounds stigmasterol alone in mice model. Furthermore, the methanol extract of adlay hull could modulate the steroidogenic enzyme activities and decrease the plasma estradiol and progesterone levels [106]. These results indicated that the adlay extracts possess the ability to reduce the growth of uterine fibroids, probably by interfering with the endocrine signaling.

### 4.4. Vitamin D

Vitamin D, a fat-soluble vitamin series, modulates several actions in the human body through its steroid compound properties [107]. The main source of vitamin D is sunlight exposure. Briefly, the radiation from sunlight causes the skin to produce vitamin D, which is transported to the liver and then kidneys to produce the active hormonal form calcitriol (1,25-dihydroxy vitamin D) [108]. Deficiency of serum vitamin D and the lower levels of Vitamin D receptor (VDR) expression in UF tumors were commonly seen in women with UFs [109,110,111], highlighting the potential modulation properties of vitamin D in UF patients. Several studies highlighted several anti-UF effects of vitamin D. Vitamin D treatment could effectively decrease ECM-related protein expression [112]. Furthermore, the toxicity of vitamin D was relatively rare; according to the vitamin D supplementation guideline, up to 50,000 IU per week was still tolerable and adequate for the vitamin D-deficient population [113]. In a previous study, HuLM cells treated with 1,25-Dihydroxyvitamin D3 showed significantly decreased progesterone receptor (PR)-A and PR-B protein expression. Consequently, 1,25(OH)_2_D3’s proliferation inhibition ability may be related to the PR modulation effect [114]. Besides the potential of vitamin D as a medical treatment, it also showed a synergistic effect when combined with UPA SPRM. Vitamin D enhanced the anti-proliferation, apoptotic and anti-fibrotic effect of UPA by reducing CyclinD1, PCNA, collagen type1, TGF-β3, and Bcl2 protein expression levels [115], showing that vitamin D may play a critical role in UF treatment and may act as adjuvant therapy in SPRM treatment. Several clinical trials explored the anti-UF effects of vitamin D all over the world [116,117], but this is beyond the scope of the current review article.

### 4.5. Traditional Herbal Medicine—Curcumae Rhizoma-Sparganii Rhizoma (CRSR)

Traditional herbal medicine has been widely used in UFs treatments [118] to decrease the consumption of medication or the surgical treatment percentage. The combination of Curcumae Rhizoma and Sparganii Rhizoma (CRSR) showed high anti-tumor [119], anti-oxidative [120], and anti-angiogenic effects [121]. A study involved treating rats with CRSR (at a dose of 6.67 g/kg for 5 weeks) and the treatment led to a significant decrease in serum estradiol and progesterone levels, likely due to the inhibition of estrogen and progesterone receptors. Additionally, the treatment improved the swelling of nodules and cysts in the uterus induced by diethylstilbestrol and progesterone, suggesting a potential benefit for UFs [122]. A recent study has shown that the CRSR modulates the cell proliferation effect and could decrease the proliferative signaling pathway Akt/MEK/ERK and extracellular matrix-related protein expression [123], indicating that CRSR has the ability to inhibit proliferation and extracellular matrix accumulation, and has the potential to modulate progesterone receptors.

### 4.6. Epigallocatechin Gallate (EGCG)

Green tea is shown to have a beneficial effect on health. Without the fermentation step, in order to keep the polyphenols component preserved, its active ingredient, EGCG, maintains its anti-proliferation and anti-metastasis effects [124]. EGCG showed capabilities on the inhibition of cyclin-dependent kinases (CDKs), induction of the cell apoptosis process, and the reduction of matrix metalloproteinases [125]. In an in vitro study, EGCG could increase the Bax and decrease the Bcl-2 expression via the inhibition of the survival-related and NFκB-dependent pathway [126]. Moreover, in a human study, treatment with EGCG (800 mg/day) for 4 months significantly reduced the severity of fibroid symptoms and improved the quality of life without any side effects [109]. The inhibition effect of EGCG may be related to several factors, including serum estrogen, progesterone, oxidative stress and even dietary intake [127]. In cancer research, EGCG has been proved to have an anti-proliferative effect via the inhibition of ER-α and PR expression [128]. In breast cancer, EGCG has also shown the ability to reduce PR expression and may be used in combination with SPRM [129]. Recent studies showed the promising effects of a green tea and vitamin D combination against UFs, where tumor size significantly decreased in the treatment group compared to the control group with QoL improvement [130,131]. There is a need for further research to confirm EGCG’s effect on modulating progesterone signaling in UFs.

### 4.7. Curcumin

Curcumin is an active component found in Curcuma longa, and is used as a food additive, cosmetic ingredient, and supplement due to its multiple health benefits, including anti-tumor, anti-inflammatory, and anti-fibrosis effects [132,133,134]. In vitro studies have shown that curcumin, at a concentration of 20 μM, significantly decreases cell proliferation. It induces apoptosis by activating caspase 3 and 9 expressions and reducing extracellular matrix-related protein expression [135,136]. Previous research has shown that curcumin has the ability to bind to progesterone receptors in semen, and reduce medroxyprogesterone acetate (MPA)-induced VEGF secretion in breast cancer cells containing estrogen and progesterone receptors [137,138,139]. However, evidence regarding curcumin’s modulation of progesterone receptors related to UFs is currently lacking.

## 5. Conclusions and Perspectives

Progesterone signaling plays a crucial role in the development and growth of UFs. Progesterone promotes the proliferation of UF cells by activating several signaling pathways genetically and epigenetically, including the Akt/MEK/ERK pathway. The expression of progesterone receptors is significantly higher in UFs tissues compared to normal myometrium. Therefore, targeting the progesterone signaling pathway may be a potential therapeutic strategy for treating UFs. Various drugs that inhibit progesterone signaling, such as SPRMs, have been developed and are currently being evaluated in clinical trials. However, further studies are needed to fully understand the mechanism of progesterone signaling in UFs and to develop more effective treatment options. In addition, there is a need for a better understanding of the interplay between various signaling pathways and their role in the development and growth of UFs. For instance, recent studies have shown that there is a crosstalk between the progesterone and estrogen signaling pathways, suggesting that both hormones may contribute to the development of UFs. Therefore, a more comprehensive approach that targets multiple signaling pathways may be required for the effective treatment of UFs. Consumption of natural compounds as potential anti-UFs treatments is promising, since these compounds can be used on a long-term basis. This is especially beneficial for women pursuing pregnancy while undertaking treatment, since other pharmacological treatments such as SPRMs are not fertility friendly. However, further research is needed in this regard.

## Figures and Tables

**Figure 1 cells-12-01117-f001:**
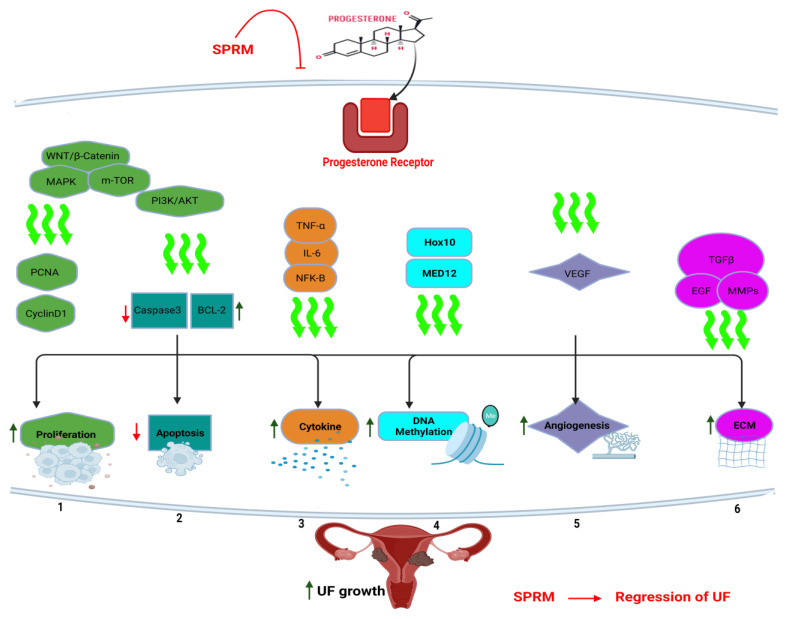
Progesterone signaling in uterine fibroid pathogenesis. Progesterone binds to its own receptor (PR), with subsequent activation of several signaling pathways including (wingless-related integration site) WNT/β-catenin pathway, mitogen-activated protein kinases (MAPK), and phosphoinositide 3-kinase (PI3K)/Ak strain transforming (AKT)/mammalian target of rapamycin (mTOR). This resulted in activation of cell proliferation, apoptosis inhibition, cytokine release, angiogenesis induction, and extracellular matrix (ECM) accumulation towards uterine fibroid (UF) tumor growth. Selective progesterone receptor modulators (SPRM) such as ulipristal acetate can interfere with progesterone signaling and induce tumor regression. PCNA: proliferating cell nuclear antigen. TNF-α: tumor necrosis factor alpha, IL-6: interleukin 6, NF-kB: nuclear factor kappa B, MED12: mediator complex subunit 12, Bcl- 2: B-cell lymphoma-2, TGF-β: transforming growth factor beta, MMP: matrix metalloproteinase, EGF: epidermal growth factor.

## Data Availability

Not applicable.

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
