# Peer review of "Progesterone Signaling and Uterine Fibroid Pathogenesis; Molecular Mechanisms and Potential Therapeutics"

_cells, 2023, doi:10.3390/cells12081117_

Round 1
Reviewer 1 Report
The authors have provided a well-rounded review of the relationship between progesterone and uterine fibroid pathogenesis and discussion of the effects of treatment with selective progesterone receptor modulators and natural compounds on fibroids. A few concerns were identified.
Major
1. A discussion of classical and non-classical progesterone signaling, including nuclear-PGR, membrane-PGR, and PGRMC1/2 should be included in section 2 on progesterone signaling. Additionally, the PGRs in figure 1 appear to be located in a membrane, which is confusing.
Minor
1. Line 36: citation #1 is a review, because the preceding sentence refers to “recent findings” the authors should site the primary literature.
2. Figure 1: figure legend should indicate that letters correspond to specific mechanisms in the body of the text.
Author Response
The authors have provided a well-rounded review of the relationship between progesterone and uterine fibroid pathogenesis and discussion of the effects of treatment with selective progesterone receptor modulators and natural compounds on fibroids. A few concerns were identified..
Response: We would like to thank the reviewer for these positive comments.
- A discussion of classical and non-classical progesterone signaling, including nuclear-PGR, membrane-PGR, and PGRMC1/2 should be included in section 2 on progesterone signaling. Additionally, the PGRs in figure 1 appear to be located in a membrane, which is confusing
Response: We thank the reviewer for this comment. In response to the reviewer’s suggestion, we have discussed this part and edited the figure.
- Minor
- Line 36: citation #1 is a review, because the preceding sentence refers to “recent findings” the authors should site the primary literature.
- Figure 1: Figure legend should indicate that letters correspond to specific mechanisms in the body of the text.
Response: We thank the reviewer for this comment. We edited the manuscript in response to the reviewer’s recommendation.
Reviewer 2 Report
The paper focuses on signalling in pathogenesis and uterine fibroid development. The authors describe the influence of numerous pathways on the induction of proliferation, apoptosis, regulation of cytokine (including TNF-α and production of interleukins), DNA methylation and angiogenesis.
The authors also describe two types of treatment affecting the development of uterine fibroids: progesterone receptors modulators eg. UPA, and natural compounds that positively affect progesterone signalling – berberine, vitamin D, curcumin.
There are 94 references elucidate the chosen topic and are well placed in the text.
· I noted one spelling error in the text – line 79 – unnecessary ‘o’ = progesterone o upregulated
I believe the review contains the necessary information providing the background and context. And the figure supports and clarifies the issue discussed.
Author Response
Reviewer 2:
The paper focuses on signaling in pathogenesis and uterine fibroid development. The authors describe the influence of numerous pathways on the induction of proliferation, apoptosis, regulation of cytokine (including TNF-α and production of interleukins), DNA methylation and angiogenesis.
The authors also describe two types of treatment affecting the development of uterine fibroids: progesterone receptors modulators eg. UPA, and natural compounds that positively affect progesterone signalling – berberine, vitamin D, curcumin. I believe the review contains the necessary information providing the background and context. And the figure supports and clarifies the issue discussed.
Response: We would like to thank the reviewer for these positive comments.
I noted one spelling error in the text – line 79 – unnecessary ‘o’ = progesterone o upregulated
Response: We thank the reviewer for this comment. We edited the manuscript in response to the reviewer’s recommendation.
Reviewer 3 Report
This is a very informative manuscript describing the impact of progesterone signaling on uterine fibroid pathogenesis. I have many suggestions that will increase readability for non-UF-specialized readers. Figure 1 is useful and informative. Section 4, detailing phytoprogestins, is very interesting. However, the manuscript could use thorough copyediting to correct many incomplete sentences and typos.
Introduction: could the authors include a paragraph on the pathogenesis of UFs, such as how they promote infertility and heavy menstrual bleeding, etc?
Section 2 should be divided into several paragraphs, each delineated by the bolded progesterone signaling functions A-F. Sections can be expanded, particularly section B on apoptosis suppression.
Line 68: could the authors describe what “fibroid side population cells” are?
Line 92: it is unclear the significance of the E2/P4 ratio, could the authors describe the significance?
Line 121: define acronym AUB
Line 132-133: minor point, but the Cochrane Review from 2017 used 5 studies about mifepristone, 4 on UPA, and 2 on asoprisnil, not 7/5/2 as stated in this manuscript.
Line 142: what is the significance of leuprolide acetate as a comparison? Could the authors describe why this is a relevant comparison? (Doing it here will also help when leuprolide acetate is referenced further down the manuscript, i.e. line 186.)
Line 150: the study using XFZY decoction is presented in this manuscript as showing reduction in UF volume, estradiol, LH, FSH, and P4, but how many of these reductions are due to XFZY and how many to mifepristone alone? If the original study compares placebo to XFZY to mifepristone to XFZY+mifepristone, could the authors differentiate in this manuscript which effects are due to mifepristone and which are due to XFZY? If not, a statement listing the inability to separate XFZY (which is not a SPRM) from mifepristone (an SPRM) should be provided in this paragraph.
Line 192: Similar to describing leuprolide acetate, could the authors briefly introduce norethisterone acetate and why it was included in the cited study as a comparison for UPA?
Lines 193-194: define acronyms TEAE and PAEC
Line 323: define acronym AHE
Line 406-408: please cite recent studies
Author Response
Reviewer 3:
This is a very informative manuscript describing the impact of progesterone signaling on uterine fibroid pathogenesis. I have many suggestions that will increase readability for non-UF-specialized readers. Figure 1 is useful and informative. Section 4, detailing phytoprogestins, is very interesting. However, the manuscript could use thorough copyediting to correct many incomplete sentences and typos.
Response: We would like to thank the reviewer for these positive comments. Per reviewer suggestion, the manuscript was extensively revised for grammar by native speaker.
- Introduction: could the authors include a paragraph on the pathogenesis of UFs, such as how they promote infertility and heavy menstrual bleeding, etc?
Response: We thank the reviewer for this comment. We edited the manuscript in response to the reviewer’s recommendation and added a paragraph regarding UF pathogenesis and underlying mechanisms of its negative impact on infertility and HMB.
- Section 2 should be divided into several paragraphs, each delineated by the bolded progesterone signaling functions A-F. Sections can be expanded, particularly section B on apoptosis suppression.
Response: We thank the reviewer for this comment. We edited the manuscript in response to the reviewer’s recommendation and separated section 2 into several paragraphs with ore details about progesterone effect on UF pathogenesis.
- Line 68: could the authors describe what “fibroid side population cells” are?
Response: We thank the reviewer for this comment. We edited the manuscript in response to the reviewer’s recommendation and added a paragrapg regarding side population.
- Line 92: it is unclear the significance of the E2/P4 ratio, could the authors describe the significance?
Response: We thank the reviewer for this comment. We edited the manuscript in response to the reviewer’s recommendation and explained the significance of E2/P4.
- Line 121: define acronym AUB
- Line 132-133: minor point, but the Cochrane Review from 2017 used 5 studies about mifepristone, 4 on UPA, and 2 on asoprisnil, not 7/5/2 as stated in this manuscript.
- Lines 193-194: define acronyms TEAE and PAEC
- Line 323: define acronym AHE
- Line 406-408: please cite recent studies
Response: We thank the reviewer for this comment. We edited the manuscript in response to the reviewer’s recommendation and defined the mentioned abbreviations..
- Line 142: what is the significance of leuprolide acetate as a comparison? Could the authors describe why this is a relevant comparison? (Doing it here will also help when leuprolide acetate is referenced further down the manuscript, i.e. line 186.)
Response: We thank the reviewer for this comment. We edited the manuscript in response to the reviewer’s recommendation and explained the leuprolide acetate as a comparison, since it was back then the only approved pharmacologic medication for women with UFs to improve the anemia before surgery.
- Line 150: the study using XFZY decoction is presented in this manuscript as showing reduction in UF volume, estradiol, LH, FSH, and P4, but how many of these reductions are due to XFZY and how many to mifepristone alone? If the original study compares placebo to XFZY to mifepristone to XFZY+mifepristone, could the authors differentiate in this manuscript which effects are due to mifepristone and which are due to XFZY? If not, a statement listing the inability to separate XFZY (which is not a SPRM) from mifepristone (an SPRM) should be provided in this paragraph.
Response: We thank the reviewer for this comment. We edited the manuscript in response to the reviewer’s recommendation and explained the comparison was between combination treatment and mifepristone alone.
- Line 192: Similar to describing leuprolide acetate, could the authors briefly introduce norethisterone acetate and why it was included in the cited study as a comparison for UPA?
Response: We thank the reviewer for this comment. We edited the manuscript in response to the reviewer’s recommendation and explained the rationale to utility of NETA in the study.
Reviewer 4 Report
This review examines the literature considering the role and mechanisms of progesterone signaling in the pathogenesis of uterine fibroids (UFs) along with the therapeutic benefits of drugs including selective progesterone receptor modulators (SPRMs) and natural compounds that modulate progesterone signaling in UFs. Overall, the review is comprehensive, relevant and useful.
While the information presented in this report provides a useful overview of the literature in this area, it would benefit significantly from a careful English editing review to improve the flow to make it an easier read.
Following is a partial list of editorial changes needed.
Line 40. “Progesterone in interrelated and involves mediating receptors…” Perhaps, The action of progesterone is complex and involves interactions with progesterone isoforms, transcription factors…”
Line 43. Change “tissues’ sensitivity” to “tissue sensitivity”
Line 43. ‘Still, there are conflicting results…” Perhaps Conflicting results remain regarding the role potential stimulatory or inhibitory effects progesterone in UF development as the mechanisms of progesterone action are not fully understood.
Line 49. Delete “review”
Line 54. “Several researches… Fugimoto et al. described…” “Several researches” is unclear and sentence ends with a single reference.
Line 63-64. Remove “…a signaling pathway known as the..”
Line 79-80. Remove the letter o after progesterone, and at “it” after meanwhile.
Line 90. Delete “Aberrant regulation of” as it changes the meaning of the sentence.
Line 109-110. The meaning of “a discrepancy between MMSC and UFSC organoids” is unclear
Line 121 “…they were investigated for against UF, AUB” meaning of “for against” is unclear; Define AUB
Line 117. I think the authors mean comprised rather than “compromised”
Line 143. Change “symptomatic” to symptoms.
Line 154. Move “considerably” to Line 155 after “fibroid volume”
Line 156. Change “The optimal dose of mifepristone was established between 2.5 and 5 mg/day for 3 or 6 months” to “The optimal dose of between 2.5 and 5 mg/day mifepristone for a period of 3 to 6 months was established.”
Line 193&194. Define TEAEs and PAECs
Line 216 …”it is still not understood [41]” What is still not understood?
Line 221-224. The meaning of these two sentences is unclear.
Line 246-247. This sentence is unclear.
Line 248. What is “Amenorrhea status?”
Line 252. Change “were presented” to “were present”.
Lines 258-260. “The problem with this SPRM is that trials examining its utility in uterine fibroid therapy were halted…” Actually, the problem with this SPRM were likely due to cases of toxicity…
Paragraph beginning Line 262. Rewrite this sentence as is does not make sense. Actually the entire paragraph should be rewritten.
Line 288. Change ,,,”are conducted…” to “have been conducted.”
Line 296-299. Convert this sentence into two sentences.
Line 313-315. Rewrite this sentence as the meaning is unclear.
Line 337. Rewrite sentence. “…promising VitaminD’s effect on UFs…
Author Response
Reviewer 4:
This review examines the literature considering the role and mechanisms of progesterone signaling in the pathogenesis of uterine fibroids (UFs) along with the therapeutic benefits of drugs including selective progesterone receptor modulators (SPRMs) and natural compounds that modulate progesterone signaling in UFs. Overall, the review is comprehensive, relevant and useful.
Response: We would like to thank the reviewer for these positive comments.
While the information presented in this report provides a useful overview of the literature in this area, it would benefit significantly from a careful English editing review to improve the flow to make it an easier read.
Response: We thank the reviewer for this comment. The manuscript was extensively reviewed for language editing by a native speaker.
- Line 40. “Progesterone in interrelated and involves mediating receptors…” Perhaps, The action of progesterone is complex and involves interactions with progesterone isoforms, transcription factors…”
- Line 43. Change “tissues’ sensitivity” to “tissue sensitivity”
- Line 43. ‘Still, there are conflicting results…” Perhaps Conflicting results remain regarding the role potential stimulatory or inhibitory effects progesterone in UF development as the mechanisms of progesterone action are not fully understood.
- Line 49. Delete “review”
- Line 54. “Several researches… Fugimoto et al. described…” “Several researches” is unclear and sentence ends with a single reference.
- Line 63-64. Remove “…a signaling pathway known as the..”
- Line 79-80. Remove the letter o after progesterone, and at “it” after meanwhile.
- Line 90. Delete “Aberrant regulation of” as it changes the meaning of the sentence.
- Line 109-110. The meaning of “a discrepancy between MMSC and UFSC organoids” is unclear
- Line 121 “…they were investigated for against UF, AUB” meaning of “for against” is unclear; Define AUB
- Line 117. I think the authors mean comprised rather than “compromised”
- Line 143. Change “symptomatic” to symptoms.
- Line 154. Move “considerably” to Line 155 after “fibroid volume”
- Line 156. Change “The optimal dose of mifepristone was established between 2.5 and 5 mg/day for 3 or 6 months” to “The optimal dose of between 2.5 and 5 mg/day mifepristone for a period of 3 to 6 months was established.”
- Line 193&194. Define TEAEs and PAECs
- Line 216 …”it is still not understood [41]” What is still not understood?
- Line 221-224. The meaning of these two sentences is unclear.
- Line 246-247. This sentence is unclear.
- Line 248. What is “Amenorrhea status?”
- Line 252. Change “were presented” to “were present”.
- Lines 258-260. “The problem with this SPRM is that trials examining its utility in uterine fibroid therapy were halted…” Actually, the problem with this SPRM were likely due to cases of toxicity…
- Paragraph beginning Line 262. Rewrite this sentence as is does not make sense. Actually the entire paragraph should be rewritten.
- Line 288. Change ,,,”are conducted…” to “have been conducted.”
- Line 296-299. Convert this sentence into two sentences.
- Line 313-315. Rewrite this sentence as the meaning is unclear.
- Line 337. Rewrite sentence. “…promising VitaminD’s effect on UFs
Response: We thank the reviewer for these useful comments and his/her thorough revision. We edited the manuscript in response to the reviewer’s recommendation.
Round 2
Reviewer 4 Report
The authors have done a good job revising the manuscript.